# CARE: CONFIDENCE-AWARE REASONING FOR RELIABLE MEDICAL-VQA

## ABSTRACT

Multimodal Large Language Models (MLLMs) have made significant progress in the medical field, yet their insufficient diagnostic reliability remains a major barrier to clinical application. To address this issue, we propose **CARE**—a novel MLLM for the medical Visual Question Answering (VQA) task, which integrates Chain-Of-Thought (CoT) reasoning and confidence awareness into its training. CARE achieves reliable diagnosis through the following approaches: First, it employs CoT to simulate the diagnostic reasoning process of physicians during Supervised Fine-Tuning (SFT). Second, it incorporates confidence estimation into the reward function of Reinforcement Fine-Tuning (RFT), significantly enhancing both answer accuracy and reasoning trustworthiness. Experimental results demonstrate that CARE consistently outperforms existing methods across multiple Medical-VQA benchmarks and exhibits strong generalization capabilities in diverse medical scenarios, which confirm that CARE not only substantially improves task accuracy but also enhances model reliability, while delivering answers with superior interpretability.

## 1 INTRODUCTION

In recent years, Multimodal Large Language Models (MLLMs) have made groundbreaking advancements in the field of computer vision, demonstrating outstanding performance in tasks such as image captioning, Visual Question Answering (VQA), and video understanding. This successful paradigm is now extending to specialized domains, particularly in healthcare, where medical MLLMs Li et al. (2024a), trained with expert medical knowledge, are driving the development of general-purpose medical AI. The current mainstream approach involves Supervised Fine-Tuning (SFT) on carefully curated multimodal medical instruction datasets, a method proven effective for handling complex tasks in real clinical scenarios. However, the direct prediction-based learning mechanism inherent in this paradigm has notable limitations: when used as a diagnostic aid for physicians, issues with reliability are becoming increasingly prominent. More importantly, the "black-box" decision-making model does not offer explainable diagnostic reasoning or inference processes, posing potential medical risks if physicians excessively rely on its diagnostic suggestions. Therefore, there is an urgent need to develop the next generation of medical MLLMs that can handle complex clinical reasoning tasks while providing verifiable diagnostic logic and reliable decision-making mechanisms.

Against this backdrop, the latest advancements in reinforcement learning offer new insights. Reinforcement Fine-Tuning (RFT) has become a hot research topic, particularly methods that combine Verifiable Rewards Reinforcement Learning (RLVR) Lambert et al. (2025) frameworks with Group Relative Policy Optimization (GRPO) Shao et al. (2024) algorithms, significantly enhancing the model's reasoning capabilities. Inspired by this, several vertical fields have launched R1 series models, with medical R1-MLLMs Lai et al. (2025) being applied in Medical-VQA scenarios. However, practical applications have revealed issues such as shallow thinking and hallucination generation in these models, with their reliability still requiring thorough validation. More critically, existing methods primarily rely on multiple-choice questions for training and evaluation, lacking the ability to handle real clinical cases, especially open-ended medical diagnosis scenarios. This raises a critical question: how can we achieve reliable reasoning of medical MLLMs in real clinical environments through high-quality data construction and optimization algorithms? At the same time, research on the alignment and calibration of LLMs and MLLMs has provided important references for improving

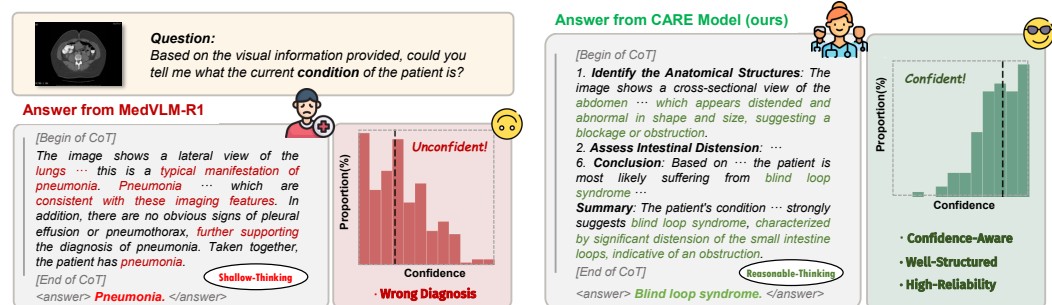

Figure 1: Illustrates a comparative case study between MedVLM-R1 and our proposed **CARE** model. The former's low reliability, stemming from its shallow reasoning, often leads to misdiagnosis, whereas the latter, empowered by the framework based on Confidence-aware, engages in deep thinking to deliver more accurate, reliable, and high-interpretable results.

model reliability. Recent works based on confidence sampling have explored feasible paths to enhance safety and reliability while maintaining high efficiency and accuracy. Although specific methods vary, these studies are all dedicated to addressing the critical issue of AI reliability. Notably, the recently released ultra-challenging benchmark test, "Humanity's Last Exam," Phan et al. (2025) revealed the limitations of existing evaluation systems: mainstream flagship MLLMs performed poorly on this test, exhibiting over-confidence that did not match their actual capabilities. This phenomenon underscores the need for reliable AI systems, a challenge that is especially critical in the high-stakes medical field, where tolerance for error is minimal. We aim for medical MLLMs to maintain a positive correlation between task accuracy and output confidence.

To address these challenges, we propose **CARE**—a **C**onfidence-**A**ware medical **RE**asoning MLLM. Specifically, our research contributions are as follows: First, based on existing Medical-VQA samples, we construct a medical reasoning dataset with high-quality structured Chain-Of-Thought (CoT), enabling the model not only to provide diagnostic conclusions but also to offer clear and reliable diagnostic reasoning processes, akin to senior radiologists or pathologists. Second, inspired by the Deepseek-R1 model, we adopt a RFT strategy based on the GRPO algorithm and innovatively design a Confidence-Aware Rewarding (CAR) strategy to achieve collaborative optimization of answer accuracy and reasoning reliability. Finally, we conducted comprehensive experimental evaluations on four mainstream Medical-VQA benchmarks, covering both open-ended and closed-ended question scenarios. As shown in Figure 1, existing medical reasoning models (e.g., MedVLM-R1) suffer from insufficient reasoning reliability, often exhibiting shallow and less rigorous reasoning processes, along with lower diagnostic accuracy and confidence. In contrast, the proposed CARE model not only incorporates an anthropomorphic and structured reasoning framework with reliable diagnostic summaries but also provides more comprehensive key information while reducing misjudgments. Subsequent experimental results and case analyses demonstrate that the CARE model significantly outperforms existing methods in both diagnostic accuracy and reasoning reliability, underscoring its strong potential as a clinical decision support system.

In summary, our contributions are as follows:

- **Confidence-aware medical reasoning framework**: Building on the advantages of the R1 model, we innovatively combine medical diagnostic Chain-Of-Thought with Reinforcement Fine-Tuning, proposing a confidence-aware reasoning strategy that enhances reasoning reliability while ensuring answer accuracy.

- **Collaborative reasoning data construction for medical scenarios**: Based on existing Medical-VQA datasets, we design a specialized method for constructing medical reasoning datasets, explicitly providing diagnostic reasoning steps and case summaries to effectively support high-reliable medical applications.

- **Optimal performance across various medical scenarios**: By applying different training strategies for various medical scenarios, we achieve significant performance improvements across multiple VQA benchmarks, proving the robustness and generalization ability of our method.

This research not only advances the development of medical MLLMs but also provides a new technical pathway for building reliable and explainable clinical decision support systems, making a significant contribution to the responsible application of AI in healthcare.

## 2 RELATED WORKS

### 2.1 MEDICAL MLLMS

With the tremendous success of Large Vision-Language Models (LVLMs) in general domains Li et al. (2024b); Alayrac et al. (2022); Zhu et al. (2023), a series of medical MLLMs Li et al. (2024a); Moor et al. (2023); Alkhaldi et al. (2024) have been developed based on them to address various clinical challenges, such as medical visual question answering, report generation, and diagnostic assistance Ye & Tang (2025). Historically, medical MLLMs can be categorized into two types: non-VLM-based Zhang et al. (2023); Li et al. (2022) and VLM-based Li et al. (2024a); Moor et al. (2023). Traditional non-VLM architectures often separate visual and language models or combine them only superficially, lacking end-to-end joint training and thus resulting in limited cross-modal fusion. In contrast, VLM architectures, with a powerful language model at their core, are adapted to directly parse raw images, achieving deep and unified visual-language reasoning. Although these models have shown impressive results, they often lack complex reasoning capabilities, which limits their generalization and clinical applicability. Consequently, a new generation of VLM-based Medical MLLMs has emerged Sun et al. (2025); Lai et al. (2025); Pan et al. (2025). These models leverage more realistic, reasoning-focused clinical data and more advanced training paradigms, pushing medical AI to a new frontier. However, a persistent challenge is the inability to judge the validity and reliability of the model's reasoning paths and final answers based solely on external metrics.

### 2.2 REINFORCEMENT FINE-TUNING (RFT)

Previously, Supervised Fine-Tuning (SFT) using Chain-Of-Thought (CoT) data was the mainstream method for enhancing the reasoning abilities of LLMs and MLLMs. However, the rise of Reinforcement Fine-Tuning (RFT) has gradually established it as a new paradigm for improving the reasoning capabilities of large models Li et al. (2025); Xu et al. (2025). This shift is attributed to the inherent advantage of RL methods in achieving a better balance between "exploration and exploitation." Unlike SFT models that passively fit existing knowledge, RFT enables models to receive direct feedback from verifiable results, thereby dynamically updating their reasoning paths and achieving superior performance with less training data Chu et al. (2025). Early research on RFT Trung et al. (2024); Zhang et al. (2024) primarily focused on its application in solving mathematical problems Shao et al. (2024) and code generation tasks Hui et al. (2024) within LLMs, with its generated high-quality reasoning steps gaining significant attention. With major breakthroughs like OpenAI's o1 OpenAI et al. (2024) and DeepSeek's R1 DeepSeek-AI et al. (2025), a large number of reasoning models aimed at solving a wide range of tasks have emerged. Recently, RFT has been extended to MLLMs Liu et al. (2025); Tan et al. (2025); Zheng et al. (2025), enabling powerful reasoning capabilities that integrate the visual modality, including applications in the medical field. As related work continues to emerge, the research community is also continuously exploring the trade-off between the performance and computational cost of RFT models Wang et al. (2025); Zhu et al. (2025); Sui et al. (2025). Furthermore, the reliability and ethical issues of AI, especially in high-risk domains like healthcare, remain critical topics. To address these challenges, we propose a medical reasoning model based on a confidence-aware RFT framework, taking a significant step toward creating robust and trustworthy medical AI.

### 2.3 CONFIDENCE CALIBRATION

Confidence, as a key metric for evaluating model outputs, is often used to enhance model alignment and calibration Geng et al. (2023); Wen et al. (2024); Liu et al. (2023). The "overconfidence" phenomenon in large models is a well-known concern, which has led to numerous methods that leverage confidence scores to constrain models, making them safer and more "honest." Kang et al. (2025) In these methods, the acquisition of confidence scores can be categorized into **white-box** Savage et al. (2024); Wang et al. (2024) and **black-box** approaches Xiong et al. (2023); Tian et al. (2023); Ni et al. (2024); Tao et al. (2024); Wada et al. (2024). **White-box** methods are applicable

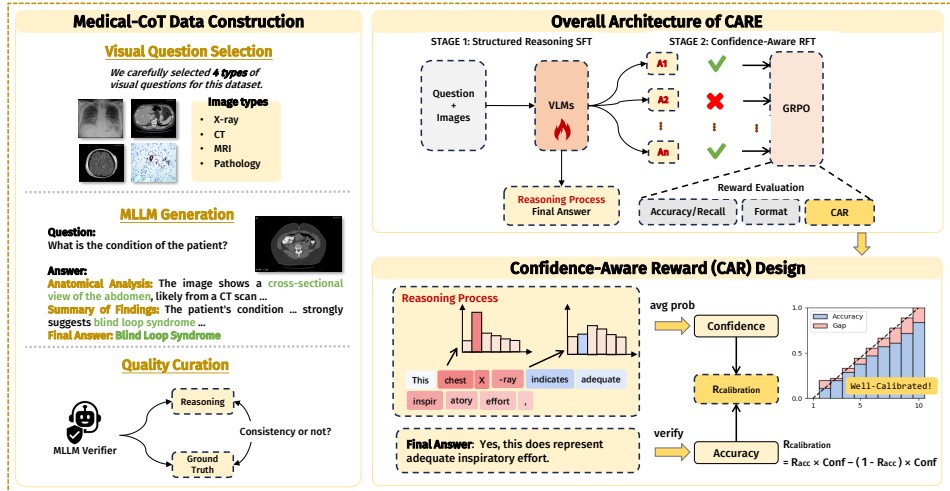

Figure 2: Overview of our method. We first generate structured analyses and answers for medical images using MLLM and validate them to create the high-quality Medical-CoT data. Then, the **CARE** model is trained in two stages: SFT in Stage 1, followed by deep RL optimization with a Confidence-Aware Reward (CAR) mechanism to enhance diagnosis accuracy and reliability.

to open-source models, deriving confidence values by analyzing the model's output probability distribution and selecting appropriate quantization techniques. Conversely, **black-box** methods require the model to explicitly state its confidence level during the sampling process, constraining this value within a predefined range through prompting. However, a core limitation is that the confidence score itself is an unsupervised signal, making it difficult to directly integrate into a reward function to enhance reasoning reliability. Our **CARE** model breaks through this barrier, allowing confidence to directly influence the training dynamics. This process strengthens the intrinsic correlation between the model's expressed confidence and the actual accuracy of its predictions, thereby making the model more aligned and reliable.

## 3 METHOD

### 3.1 OVERVIEW

Figure 2 illustrates the overall process of our approach. We first use MLLM to generate structured analyses and answers for various medical images such as X-rays and CT scans, and, through rigorous validation by auxiliary models, construct a high-quality Medical-CoT dataset. Next, we adopt a two-stage architecture to train the CARE model: In the first stage, the model learns to perform basic structured reasoning and provide answers; in the second stage, deep optimization is carried out through a core **Confidence-Aware Reward (CAR)** mechanism. This mechanism evaluates the model's confidence in its own reasoning and, by combining the actual accuracy of the answer, generates a calibrated reward signal. This effectively guides the model to not only make correct diagnoses but also provide an accurate assessment of the reliability of its judgments.

### 3.2 MEDICAL CoT DATA CONSTRUCTION

Before training, we constructed high-quality Chain-Of-Thought (CoT) data based on existing Medical-VQA datasets for CARE. Our approach focuses on the functional role of CoT data as a training signal rather than treating each reasoning path as a standalone, clinically perfect diagnosis. Inspired by Wang et al., we innovatively adopted a reverse-thinking strategy, using the base model $\pi_\theta$ itself to automatically generate diverse reasoning paths. Given visual context $V$, text query $Q$, and ground truth answer $GT$, we guide $\pi_\theta$ to generate corresponding intermediate reasoning steps through carefully designed prompts.

$$\mathcal{T} \leftarrow \pi_\theta(V, Q, GT) \tag{1}$$

Considering the functional purpose of this data, direct manual validation by medical experts for each reasoning path is not essential. The objective is not to ensure each path is clinically flawless, but that it constitutes a plausible and logical trajectory leading to the correct answer, thereby effectively guiding the model's learning process. To this end, we implemented a pragmatic and scalable automated verification process. To ensure generation quality, we first strictly structured the reasoning paths, requiring diagnostic steps as phase identifiers at the beginning and a summarized statement at the end, enforced by a multi-round rule-checking mechanism. Subsequently, we introduced GPT-4o as an auxiliary verifier with a strict criterion: a generated reasoning path is deemed valid for training only if its final conclusion aligns perfectly with the ground truth answer $GT$. This automated check ensures that the generated CoTs are goal-oriented and logically coherent enough to arrive at the correct outcome. The ultimate validation of this data generation strategy lies not in the subjective assessment of individual training paths, but in the objective performance improvements of the final trained model, which, as our main results demonstrate, confirms the effectiveness of our approach.

## 3.3 DUAL-STAGE TRAINING OF CARE

Using the constructed medical reasoning dataset, CARE adopts a two-phase RFT strategy: SFT cold start activation and GRPO-based RL optimization. The SFT cold start phase trains the model to decompose complex tasks into logical steps.

**SFT Cold Start Phase**: This phase trains the model to break down the overall task into corresponding medical diagnostic steps. Each sample is represented as $(x, q, r, a)$, where $x$ is the input image, $q$ is the question, $r$ is the reasoning steps, and $a$ is the final answer. The training objective is to maximize the likelihood of generating $r$ and $a$ given $(x, q)$:

$$L_{\text{SFT}} = -E_{\substack{(x,q,r,a) \\ \sim \mathcal{D}}} \sum_{t=1}^{T} \log \pi_\theta(y_t | x, q, y_{<t}) \tag{2}$$

Here, $\mathcal{D}$ is the dataset, $y$ is the concatenated sequence of $r$ and $a$, and $\pi_\theta$ is the token distribution of the model. The output model of this phase serves as the initialization for the next phase of reinforcement learning, ensuring a solid foundation for RL.

**RL Phase**: Since the GRPO algorithm is resource-friendly (it does not require loading an independent value model of the same size as the policy model during training), we chose it as the optimization algorithm for the RL phase. During training, the initial version of the policy model $\pi_\theta$ is used as the reference model $\pi_{\text{ref}}$. In each training loop, we sample an image $x$ and its corresponding question $(x, q)$ from the dataset. Then, the old policy model $\pi_\theta^{\text{old}}$ generates a set of $k$ candidate outputs $o_1, o_2, \ldots, o_k$, where each output $o_i = (r_i, a_i)$ represents the reasoning trajectory and final prediction result. Next, the GRPO algorithm optimizes the policy model by maximizing the following objective function:

$$J_{GRPO}(\theta) = E\left[\frac{1}{k}\sum_{i=1}^{k} R(x, q, o_i)\right] - \beta D_{KL}(\pi_\theta \| \pi_{ref}) \tag{3}$$

Here, $R(x, q, o_i)$ is a clipped surrogate objective function based on the importance ratio and advantage estimate $A_i$. The importance ratio $\frac{\pi_\theta(o_i|v,q)}{\pi_\theta^{\text{old}}(o_i|v,q)}$ is used to measure the difference between the new and old strategies. The $D_{KL}(\pi_\theta \| \pi_{ref})$ is a KL divergence penalty term that ensures the updated model does not diverge too far from the initial reference model, stabilizing the training process. In practice, the reward function we designed scores each output $o_i$, resulting in reward scores $r_1, r_2, \ldots, r_k$. The advantage value $A_i$ is the normalized result of these reward scores:

$$A_i = \frac{R_i - \text{mean}(\{R\})}{\text{std}(\{R\})} \tag{4}$$

This advantage value reflects the quality of the corresponding output. Ultimately, reasoning paths with reward values above the mean will have a higher sampling probability during the generation process, while paths below the mean will be suppressed.

## 3.4 CONFIDENCE-AWARE REWARD (CAR) DESIGN

In our practice, the reward function used for the advantage value $A_i$ in the GRPO algorithm consists of several key components. Unlike the conventional approach in the RLVR paradigm, which only employs simple accuracy and format rewards, we innovatively introduce a confidence-aware reward mechanism. By incorporating confidence calibration into the reward function design, we significantly enhance the model's performance, reliability, and safety. Specifically, our reward function group includes three core parts: format reward, output reward, and calibration reward.

**The format reward** is responsible for verifying the structural correctness of output $o_i$, requiring the model to generate the reasoning process $r_i$ within the <think></think> tags and the final answer $a_i$ within the <answer></answer> tags:

$$R_{\text{form}}(o_i) = \begin{cases} 1, & \text{if the output format is correct,} \\ 0, & \text{otherwise.} \end{cases} \tag{5}$$

**The output reward** employs a differentiated evaluation strategy based on the question type. For closed questions, a strict exact-matching mechanism is applied (if the ground truth, GT, is fully contained in the final answer $a_i$, then 1 point is given; otherwise, 0 points). For open-ended questions, we adopted a recall-based evaluation method, focusing on the coverage of GT content in the output $o_i$:

$$R_{\text{out}}(o_i, a_i, \text{GT}) = \begin{cases} I(\text{GT} \subseteq a_i), & \text{if close-ended} \\ \text{recall}(o_i, \text{GT}), & \text{otherwise} \end{cases} \tag{6}$$

**The calibration reward** aims to measure the **consistency** of the model's output with its own knowledge. First, we quantify the confidence score for each output. We obtain the log-likelihood for each token $t_j$ in the output sequence, $\log P(t_j)$, and convert it to the corresponding probability value. Then, we calculate the average of these probability values to obtain the confidence score.

$$C(o_i) = \frac{1}{|o_i|} \sum_{j=1}^{|o_i|} P(t_j) \tag{7}$$

Next, we can compute the calibration reward $R_{\text{calib}}$:

$$R_{\text{calib}}(o_i, a_i, \text{GT}) = R_{\text{out}} \cdot C(o_i) - (1 - R_{\text{out}}) \cdot C(o_i) \tag{8}$$

This innovation establishes a quantifiable connection between model prediction accuracy and reliability. This mechanism not only reflects the model's confidence in its output but also enables the reinforcement learning framework to adjust the model's exploration strategy, driving the model to improve the robustness of its outputs while maintaining high accuracy. Ultimately, these reward components are combined to form the foundational reward score $R$ for advantage value computation, enabling precise guidance of the model's exploration strategy.

$$R = R_{\text{form}} + R_{\text{out}} + R_{\text{calib}} \tag{9}$$

## 4 EXPERIMENTS

### 4.1 EXPERIMENTAL SETTING

**Dataset.** We evaluate our CARE model on four publicly available Medical-VQA datasets:

- **VQA-RAD** Lau et al. (2018): A Medical-VQA dataset focused on radiology, containing 315 radiology images annotated by clinicians and 3,515 question-answer pairs.
- **SLAKE** Liu et al. (2021): A semantically-labeled, knowledge-enhanced Medical-VQA dataset, containing 642 images and 14,000 bilingual question-answer pairs.
- **PATH-VQA** He et al. (2020): A pathology image dataset, containing a total of 4,998 pathology images and 32,799 question-answer pairs.

Table 1: Main Result table with best performances bolded.

| Model / Dataset | VQA-RAD | | | SLAKE | | | PathVQA | | | OmniMedVQA |
|---|---|---|---|---|---|---|---|---|---|---|
| | Open | Closed | All | Open | Closed | All | Open | Closed | All | All |
| *Non-Reasoning Medical MLLMs* | | | | | | | | | | |
| HuatuoGPT-Vision-7B | 31.88 | 66.90 | 53.00 | 42.20 | 59.80 | 49.10 | 11.07 | 52.90 | 32.00 | / |
| Med-Flamingo-9B | 50.00 | 65.07 | 59.09 | 78.18 | 63.22 | 72.31 | 7.74 | 63.20 | 35.49 | 38.51 |
| HealthGPT-14B | 28.82 | 77.70 | 58.30 | 56.82 | 76.40 | 64.50 | 2.83 | 85.90 | 44.40 | / |
| Med-MoE-3.6B | 58.55 | 82.72 | 73.13 | 85.06 | 85.58 | 85.26 | 34.74 | 91.98 | 63.38 | 46.85 |
| Llava-Med-7B | 61.52 | 84.19 | 75.19 | 83.08 | 85.34 | 83.97 | 37.95 | 91.21 | 64.60 | 25.43 |
| *Reasoning Medical MLLMs* | | | | | | | | | | |
| Med-R1-3B | 40.99 | 58.09 | 51.30 | 52.58 | 70.43 | 59.58 | 14.88 | 62.49 | 38.70 | 65.34 |
| MedVLM-R1-2B | 34.14 | 65.81 | 53.24 | 41.23 | 64.66 | 50.42 | 12.80 | 64.46 | 38.65 | 76.04 |
| *Ours* | | | | | | | | | | |
| CARE-3B | 57.30 | 76.84 | 69.08 | 85.20 | 81.01 | 83.56 | 36.88 | 82.15 | 59.53 | 76.70 |
| **CARE-7B** | **61.98** | **86.40** | **76.71** | **88.13** | **86.06** | **87.32** | **42.13** | **95.54** | **68.86** | **81.53** |

- **OmniMedVQA** Hu et al. (2024): a novel comprehensive Medical-VQA benchmark, containing a total of 127995 question-answer pairs.

**Baseline Methods.** We compare the CARE model with the following two categories of state-of-the-art (SOTA) baseline models: (1) Non-Reasoning Medical MLLMs: This includes vision models pre-trained on specific medical corpora, such as HuatuoGPT-Vision-7B Chen et al. (2024), Med-Flamingo-9B Moor et al. (2023), HealthGPT-14B Lin et al. (2025), Med-MoE-3.6B Jiang et al. (2024), and Llava-Med-7B Li et al. (2024a). (2) Reasoning Medical MLLMs: This includes Med-R1-3B Lai et al. (2025) and MedVLM-R1-2B Pan et al. (2025). We measure the performance of models on the VQA benchmarks using recall for open-ended questions and accuracy for closed-ended questions. For the OmniMedVQA evaluation, we assessed the recall performance of the open-access data across the three modalities: CT, MRI, and X-ray.

## 4.2 IMPLEMENTATION DETAILS

During the training phase, we use Qwen2.5-VL-3B-Instruct Bai et al. (2025) and Qwen2.5-VL-7B-Instruct as the base models for CARE, performing full-parameter fine-tuning on a server cluster equipped with 6×A100 GPUs. The SFT phase utilizes the AdamW optimizer with a learning rate set to 1e-5, following a cosine annealing schedule. To balance training accuracy and efficiency, the RFT phase generates 4 rollouts per sample with a batch size of 2, using bfloat16 mixed precision.

## 4.3 MAIN RESULTS

**Overall performance comparisons.** To comprehensively evaluate the performance of our proposed **CARE** model, we conducted extensive experiments on four Medical-VQA benchmark datasets, comparing the performance of CARE-3B/7B with current state-of-the-art non-reasoning and reasoning medical MLLMs. As shown in Table 1, we evaluated the models' performance on the VQA-RAD, SLAKE, and PathVQA datasets across open-ended (Open), closed-ended (Closed), and overall (All) settings. Since the OmniMedVQA dataset consists entirely of multiple-choice questions, we only report its overall (All) accuracy.

In general, CARE-7B demonstrated superior performance across all settings on the four datasets. On the **VQA-RAD** dataset, CARE-7B achieved 61.98% on open-ended questions, 86.40% on closed-ended questions, and an overall accuracy of 76.71%, outperforming all competing models. Compared to the current best-performing non-reasoning model, Llava-Med-7B (with an overall accuracy of 75.19%), CARE-7B demonstrated a steady improvement, with an increase of 0.46 percentage points on open-ended tasks and 1.52 percentage points overall. Furthermore, CARE-3B achieved an overall accuracy of 69.08%, also surpassing most models with a similar parameter scale, reflecting a good balance between performance and model efficiency. On the **SLAKE** dataset, CARE-7B reached 88.13%, 86.06%, and 87.32% for open-ended, closed-ended, and overall settings, respectively. Particularly in open-ended tasks, CARE-7B showed a significant 3.07 percentage point improvement over the previous leading model, Med-MoE-3.6B (85.06%), demonstrating extremely

strong performance on medical tasks requiring deeper understanding and generation capabilities. CARE-3B also showed competitive performance on this dataset with an overall accuracy of 83.56%. On the **PathVQA** dataset, CARE-7B once again ranked first in all settings (42.13%, 95.54%, and 68.86%). Notably, in open-ended tasks, CARE-7B surpassed the second-best model, Llava-Med-7B (37.95%), by 4.18 percentage points, and its closed-ended accuracy also reached the current highest level. On the **OmniMedVQA** dataset, CARE-7B also achieved the highest overall accuracy of 81.53%, significantly outperforming other models like MedVLM-R1 (76.04%). This indicates that our model maintains excellent robustness and performance when processing medical images across different modalities and scenarios. In conclusion, the experimental results clearly demonstrate that **CARE**, particularly CARE-7B, consistently leads in terms of accuracy and reasoning reliability across the four major Medical-VQA benchmarks, fully validating the effectiveness and advanced nature of our model's design.

Table 2: Overall accuracy and ECE of **CARE** and other medical MLLMs. Smaller ECEs corresponds to more calibrated and reliable results.

| Models | VQA-RAD | | SLAKE | | PathVQA | | OmniMedVQA | |
|---|---|---|---|---|---|---|---|---|
| | Acc.↑ | ECE ↓ | Acc.↑ | ECE ↓ | Acc.↑ | ECE ↓ | Acc.↑ | ECE ↓ |
| Med-R1-3B | 51.30 | 45.07 | 59.58 | 36.90 | 38.70 | 58.38 | 65.34 | 33.62 |
| MedVLM-R1-2B | 53.24 | 44.77 | 50.42 | 47.39 | 38.65 | 58.69 | 76.04 | 22.19 |
| Med-MoE-3.6B | 73.13 | 22.13 | 85.26 | 12.64 | 63.38 | 30.18 | 46.85 | 46.58 |
| Llava-Med-7B | 75.19 | 52.36 | 83.97 | 43.73 | 64.60 | 55.94 | 25.43 | 67.28 |
| **CARE-7B** | **76.71** | **20.23** | **87.32** | **11.45** | **68.86** | **29.02** | **81.53** | **16.76** |

**Comparisons of reliability by confidence-related metrics.** In terms of reliability evaluation, this study employs the Expected Calibration Error (ECE) as a quantitative metric. This metric assesses reasoning reliability by measuring the alignment between a model's confidence and its prediction accuracy. Specifically, ECE divides prediction confidences into $M$ equal-width bins $\{B_m | m = 1, ..., M\}$ and calculates the weighted average difference between the mean accuracy and the mean confidence within each bin, thereby precisely quantifying the degree of deviation between model confidence and prediction accuracy:

$$\text{ECE} = \sum_{m=1}^{M} \frac{|B_m|}{N} |\text{acc}(B_m) - \text{conf}(B_m)| \tag{10}$$

As shown in Table 2, we compared CARE-7B with several baseline models. The experimental results strongly demonstrate that our method not only significantly enhances the model's overall performance but, more importantly, also ensures the reliability of the reasoning process. In terms of accuracy, CARE-7B achieved state-of-the-art levels on the VQA-RAD (76.71%), SLAKE (87.32%), PathVQA (68.86%), and OmniMedVQA (81.53%) datasets. In terms of reliability, its ECE scores were as low as 20.23, 11.45, 29.02, and 16.76, respectively, marking the best performance among all compared models. It is worth noting that we found existing reasoning models (such as Med-R1-3B, MedVLM-R1-2B), despite having certain reasoning capabilities, generally exhibit poor reliability, with ECE metrics at high levels (>33). In contrast, our CARE-7B model, with its confidence-aware reasoning framework, significantly surpassed them in all tests, for instance, drastically reducing the ECE on VQA-RAD from approximately 45 to 20.23 and on OmniMedVQA from over 33 to 16.76. This indicates that our method not only improves model performance but also makes its judgments and reasoning more cautious and trustworthy.

Table 3: Ablation study results for different module configurations. CA: confidence-aware reward design; RE: generated medical reasoning data.

| Setting | VQA-RAD | | SLAKE | | PathVQA | | OmniMedVQA | |
|---|---|---|---|---|---|---|---|---|
| | Acc.↑ | ECE ↓ | Acc.↑ | ECE ↓ | Acc.↑ | ECE ↓ | Acc.↑ | ECE ↓ |
| **CARE** | **76.71** | **20.23** | **87.32** | **11.45** | **68.86** | **29.02** | **81.53** | **16.76** |
| w/o RE | 58.18 | 38.13 | 82.92 | 15.75 | 61.89 | 36.54 | 63.84 | 35.38 |
| w/o CA | 62.10 | 35.53 | 85.31 | 13.25 | 66.63 | 30.85 | 70.22 | 26.78 |
| w/o CA+RE | 56.93 | 42.41 | 79.80 | 22.36 | 59.67 | 38.06 | 62.63 | 36.80 |

Table 4: Ablation study results for different training stages.

| Training Stages | VQA-RAD | | | | SLAKE | | | | PathVQA | | | | OmniMedVQA | |
|---|---|---|---|---|---|---|---|---|---|---|---|---|---|---|
| | Open ↑ | ECE ↓ | Closed ↑ | ECE ↓ | Open ↑ | ECE ↓ | Closed ↑ | ECE ↓ | Open ↑ | ECE ↓ | Closed ↑ | ECE ↓ | Acc.↑ | ECE ↓ |
| Training-Free | 48.25 | 48.45 | 65.81 | 34.13 | 51.30 | 45.58 | 68.99 | 28.65 | 15.02 | 82.33 | 66.39 | 32.57 | 75.78 | 20.40 |
| SFT | 51.98 | 46.83 | 62.87 | 35.93 | 80.31 | 18.82 | 76.68 | 22.45 | 34.92 | 63.97 | 85.72 | 13.26 | 79.49 | 18.64 |
| RL | 56.27 | 40.65 | **86.40** | **9.62** | 81.89 | 17.56 | **86.06** | **11.88** | 16.59 | 72.70 | **95.54** | **2.03** | 78.70 | 19.33 |
| SFT+RL | **61.98** | **36.34** | 68.38 | 27.49 | **88.13** | **11.18** | 77.88 | 20.93 | **42.13** | **56.05** | 86.29 | 12.09 | **81.53** | **16.76** |

## 4.4 ABLATION STUDIES

**Different Module Configurations.** As shown in Table 3, we conducted a thorough ablation study to dissect the individual contributions of CARE's key modules. The results unequivocally show that the complete model consistently achieved the best performance across all datasets. Removing the **Reasoning Data (w/o RE)** caused a significant drop in accuracy. This finding highlights the critical role of CoT data. More revealingly, removing the **Confidence-Aware module (w/o CA)** not only reduced accuracy but, more critically, severely degraded model reliability, as evidenced by a sharp increase in ECE. This indicates that the model became dangerously overconfident, providing incorrect answers with high certainty—a failure mode that is unacceptable in clinical applications. This result powerfully validates our CA module's effectiveness in calibrating model confidence. Finally, removing both modules **(w/o CA+RE)** led to a synergistic performance collapse, yielding the worst results across the board. This confirms that both components are indispensable and function in a complementary manner to achieve both high accuracy and trustworthiness.

**Different Training Stages.** As detailed in Table 4, our analysis of training stages revealed intriguing patterns for open-ended and closed-ended question formats. For **open-ended questions**, which require generating descriptive answers, the two-stage **SFT+RL** framework was clearly superior. The SFT stage provides an essential foundation by teaching the model the necessary domain knowledge, reasoning structure, and response style. The subsequent RL fine-tuning then sharpens this generative ability for greater factual accuracy and reliability, resulting in the best overall performance in both accuracy and ECE. In stark contrast, for **closed-ended questions** (e.g., Yes/No), which demand a discriminative choice rather than generation, using **only the RL** stage was far more effective, achieving the highest accuracy and the lowest ECE (e.g., 95.54% accuracy and 2.03 ECE on PathVQA). Since answer choices are predefined, skipping the SFT stage trains the model as a direct "decision-maker," optimizing its policy for logical judgment and avoiding redundant generative biases. Interestingly, on the **OmniMedVQA** benchmark—also a multiple-choice task—the combined **SFT+RL** configuration achieved the best accuracy (81.53%) and reliability (ECE 16.76). This suggests OmniMedVQA's questions are complex enough to benefit from the broad knowledge base instilled by SFT and later refined by RL, reinforcing the value of our combined approach for robust performance on challenging benchmarks.

## 5 CONCLUSION AND DISCUSSION

To address the reliability issues in medical MLLMs, we propose the Confidence-Aware REasoning model, CARE. Through high-quality medical reasoning data and a novel reward mechanism, CARE surpasses existing models on multiple Medical-VQA benchmarks. It not only improves performance, but its lower Expected Calibration Error (ECE) also indicates that it effectively alleviates the model's overconfidence problem.

We acknowledge the computational cost of creating our CoT dataset. However, we consider this a necessary, one-time investment for a crucial trade-off. In high-stakes fields like medicine, prioritizing model reliability over computational cost is essential. The significant gains in trustworthiness and performance justify this upfront effort. This reliability-centric approach is valuable for other critical domains, and we anticipate the cost will decrease as underlying technologies become more efficient.

The core contribution of this research is the proposal of a new paradigm for enhancing the reliability and interpretability of medical MLLMs. By integrating confidence into the learning process, CARE not only ensures diagnostic accuracy but also makes the reasoning process more transparent and trustworthy, providing a key technical pathway for building safe next-generation clinical decision support systems.

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

# A APPENDIX: REPRODUCIBILITY CHECKLIST

Unless specified otherwise, please answer "yes" to each question if the relevant information is described either in the paper itself or in a technical appendix with an explicit reference from the main paper. If you wish to explain an answer further, please do so in a section titled "Reproducibility Checklist" at the end of the technical appendix. This paper:

- Includes a conceptual outline and/or pseudocode description of AI methods introduced (yes/partial/no/NA) yes
- Clearly delineates statements that are opinions, hypothesis, and speculation from objective facts and results (yes/no) yes
- Provides well marked pedagogical references for less-familiare readers to gain background necessary to replicate the paper (yes/no) yes

Does this paper make theoretical contributions? (yes/no) yes

If yes, please complete the list below.

- All assumptions and restrictions are stated clearly and formally. (yes/partial/no) yes
- All novel claims are stated formally (e.g., in theorem statements). (yes/partial/no) yes
- Proofs of all novel claims are included. (yes/partial/no) yes
- Proof sketches or intuitions are given for complex and/or novel results. (yes/partial/no) yes
- Appropriate citations to theoretical tools used are given. (yes/partial/no) yes
- All theoretical claims are demonstrated empirically to hold. (yes/partial/no/NA) yes
- All experimental code used to eliminate or disprove claims is included. (yes/no/NA) yes

Does this paper rely on one or more datasets? (yes/no) yes

If yes, please complete the list below.

- A motivation is given for why the experiments are conducted on the selected datasets (yes/partial/no/NA) yes
- All novel datasets introduced in this paper are included in a data appendix. (yes/partial/no/NA) yes
- All novel datasets introduced in this paper will be made publicly available upon publication of the paper with a license that allows free usage for research purposes. (yes/partial/no/NA) yes
- All datasets drawn from the existing literature (potentially including authors' own previously published work) are accompanied by appropriate citations. (yes/no/NA) yes
- All datasets drawn from the existing literature (potentially including authors' own previously published work) are publicly available. (yes/partial/no/NA) yes
- All datasets that are not publicly available are described in detail, with explanation why publicly available alternatives are not scientifically satisficing. (yes/partial/no/NA) yes

Does this paper include computational experiments? (yes/no) yes

If yes, please complete the list below.

- This paper states the number and range of values tried per (hyper-) parameter during development of the paper, along with the criterion used for selecting the final parameter setting. (yes/partial/no/NA) partial
- Any code required for pre-processing data is included in the appendix. (yes/partial/no) yes
- All source code required for conducting and analyzing the experiments is included in a code appendix. (yes/partial/no) yes

- All source code required for conducting and analyzing the experiments will be made publicly available upon publication of the paper with a license that allows free usage for research purposes. (yes/partial/no) yes

- All source code implementing new methods have comments detailing the implementation, with references to the paper where each step comes from (yes/partial/no) partial

- If an algorithm depends on randomness, then the method used for setting seeds is described in a way sufficient to allow replication of results. (yes/partial/no/NA) yes

- This paper specifies the computing infrastructure used for running experiments (hardware and software), including GPU/CPU models; amount of memory; operating system; names and versions of relevant software libraries and frameworks. (yes/partial/no) partial

- This paper formally describes evaluation metrics used and explains the motivation for choosing these metrics. (yes/partial/no) yes

- This paper states the number of algorithm runs used to compute each reported result. (yes/no) yes

- Analysis of experiments goes beyond single-dimensional summaries of performance (e.g., average; median) to include measures of variation, confidence, or other distributional information. (yes/no) yes

- The significance of any improvement or decrease in performance is judged using appropriate statistical tests (e.g., Wilcoxon signed-rank). (yes/partial/no) yes

- This paper lists all final (hyper-)parameters used for each model/algorithm in the paper's experiments. (yes/partial/no/NA) partial

## B   APPENDIX: LLM USE

In the preparation of this manuscript, we utilized a Large Language Model (LLM) as a general-purpose writing assistant. The use of the LLM was strictly confined to improving the language and rhetoric of the text. Its role was limited to tasks such as correcting grammatical errors, rephrasing sentences for better clarity, and polishing the overall prose. The LLM did not contribute to the conceptualization of our core ideas, the experimental design, the analysis of results, or the formulation of scientific conclusions. All fundamental concepts and intellectual contributions presented in this paper are solely those of the authors, who take full responsibility for the accuracy and integrity of the content.

# C APPENDIX: SUPPLEMENTARY MATERIALS

Figure 3: We use visualized confusion matrices to investigate in detail the respective impacts of the confidence-aware (CA) and reasoning (RE) components on the accuracy of answers for open-ended and closed-ended questions across different datasets. From left to right: SLAKE, VQA-RAD, and PathVQA.

## C.1 VISUALIZATION AND ANALYSIS

To delve deeper into the core contribution of our proposed **Confidence-Aware (CA)** module to model accuracy and to analyze it in conjunction with another key component—the **Reasoning Data (RE)**—we conducted a visualization analysis across various datasets. As shown in Figure 3, the confusion matrices intuitively illustrate the performance improvement from answers corrected from incorrect to correct (green area) after introducing the module, as well as the potential impact of answers changing from correct to incorrect (orange area).

Specifically, for the **SLAKE** dataset (leftmost confusion matrix), the **CA** module demonstrates significant practical value, especially when handling complex open-ended questions, where it successfully corrected up to 22.16% of wrong answers while introducing only 4.2% new errors. This result strongly proves that the **CA** module can effectively perceive the model's own uncertainty and guide it toward more reliable judgments, thereby significantly enhancing the model's performance. For the **VQA-RAD** dataset (middle confusion matrix), the **RE** process is also a key factor in improving model performance, correcting 27.02% of errors on open-ended questions. Similar trends were observed on the **PathVQA** dataset (rightmost confusion matrix), further confirming the robust contribution of both modules.

In summary, this cross-dataset analysis confirms the critical role of our proposed **CA** and **RE** modules in enhancing both the model's accuracy and reliability. This is particularly crucial for navigating the diagnostic ambiguity inherent in many clinical scenarios. The significant gains observed across SLAKE, VQA-RAD, and PathVQA datasets, especially in handling open-ended questions with high uncertainty, establish these modules as key innovations within our system. Ultimately, this demonstrates CARE's strong potential for trustworthy application in real-world clinical practice.

## C.2 PROMPT SETTINGS

To support our framework, we designed two distinct types of prompts for different stages of the process.

The first is a "reverse-thinking" prompt, illustrated in Figure 4, which is used for generating the high-quality medical Chain-of-Thought (CoT) dataset. This prompt provides an advanced model with an existing medical image, a question, and a ground-truth answer. It then instructs the model to construct a detailed, step-by-step reasoning process that logically leads to the provided answer. To ensure the quality and consistency of the generated data, the prompt enforces specific requirements, such as a minimum word count and a concluding summary, without repeating the final answer itself.

The second prompt, shown in Figure 5, is the instruction template used during the training and evaluation of the CARE model. This prompt directs the model to first analyze the given medical

> *Based on the following medical question and image, generate a detailed thought process to explain how to derive the answer from the inputs.*
>
> *Image:* {<image></image>}
> *Question:* {original_question}
> *Answer:* {original_answer}
>
> *Requirements:*
> *1. The reasoning must be step-by-step and clearly divided into points (1., 2., 3., etc.)*
> *2. The total length must be at least 200 words*
> *3. End with a clear summary and proper punctuation (must end with '.')*
> *4. Do not output the answer, only generate the reasoning process.*

Figure 4: Demonstrates our reverse-thinking prompt to generating high-quality medical Chain-of-Thought (CoT) datasets. Leveraging existing visual question-answering (VQA) datasets containing images and QA pairs, we employ advanced models to produce high-reasoning processes, followed by rigorous format and content verification.

> *Look the given medical image carefully, and complete the tasks below.*
>
> *Your task:*
> *1. Think through the question step by step, enclose your reasoning process in <think>...</think> tags.*
> *2. Then provide the correct answer inside <answer>...</answer> tags.*
> *3. No extra information or text outside of these tags.*
>
> *<image></image>*
> *Question:* Does the picture contain kidney?

Figure 5: Our instruction prompt templates guide the CARE model to perform format-compliant reasoning on visual inputs, producing detailed thought processes and precise answers during both training and evaluation.

image and question, then externalize its reasoning process within <think></think> tags, and finally provide a concise answer within <answer></answer> tags. This structured format ensures that the model's outputs are consistent and easily parsable, separating the reasoning from the final conclusion.

## C.3 MEDICAL COT DATASET EXAMPLES

Our data generation process produces detailed, high-quality reasoning paths for both open-ended and closed-ended questions across different medical domains.

Figure 6 shows an example of an open-ended question based on a pathology image. The model is asked to characterize "hyperplasia without atypia". The generated reasoning process methodically breaks down the task into five steps: identifying key histological features, analyzing glandular structures, examining cell arrangement, contextualizing the condition, and drawing a final conclusion. This detailed analysis leads to the correct final answer, "By nests of closely packed glands".

Figure 7 provides an example of a closed-ended question related to a radiology image (a cross-sectional scan). The question is a simple verification: "Does the picture contain kidney?". The model's reasoning process again follows a logical progression, starting with an initial observation of the image type, placing it in an anatomical context, comparing the visible structures to known kidney anatomy (such as the renal pelvis and cortex), and concluding based on the evidence. This leads to the correct binary answer, "Yes".

---

**Example of Our Medical-CoT Data**

**Question:** How is hyperplasia without atypia characterized?

**Reasoning Process:**

**1. Identify the Key Features:** The image provided appears to depict a histological section of tissue, likely from a glandular structure, given the presence of gland-like structures. Hyperplasia refers to an increase in the number of cells within a tissue or organ, while atypia indicates abnormal cell morphology or behavior that could suggest malignancy.

**2. Analyze the Image for Glandular Structures:** In the image, there are distinct areas that resemble glandular structures, which are composed of epithelial cells arranged in a specific pattern. These structures are typically lined by a single layer of cells and may contain secretory products.

**3. Examine the Cell Arrangement:** The glands in the image appear to be tightly packed together, forming nests. This arrangement suggests a dense clustering of cells, which is characteristic of hyperplasia where there is an increased number of cells but they maintain their normal morphology.

**4. Consider the Context of Hyperplasia Without Atypia:** Hyperplasia without atypia means that the cells are increasing in number but do not show any signs of abnormality in their appearance or behavior. This condition is often benign and can be seen in various physiological processes such as hormonal stimulation or chronic inflammation.

**5. Conclusion from the Analysis:** Based on the image and the characteristics described, the nests of closely packed glands observed in the image are indicative of hyperplasia without atypia. This is because the glands are densely packed, suggesting an increase in cell numbers, but there is no evidence of atypia, meaning the cells retain their normal structure and function.

**Summary:** The image shows nests of closely packed glands, which is characteristic of hyperplasia without atypia, indicating an increase in cell numbers without any abnormal cell morphology or behavior.

**Final Answer:** By nests of closely packed glands.

Figure 6: Sample from our medical reasoning dataset: An **open-ended** analysis of a **pathology** case, illustrating how our model breaks down diagnostic imaging through step-by-step clinical reasoning.

---

**Example of Our Medical-CoT Data**

**Question:** Does the picture contain kidney?

**Reasoning Process:**

**1. Initial Observation:** The provided image is a cross-sectional view of an organ, likely obtained through medical imaging such as MRI or CT scan. The grayscale nature of the image suggests it is a type of medical imaging that uses contrast to highlight different tissues and structures within the body.

**2. Anatomical Context:** The shape and structure visible in the image are consistent with the appearance of a kidney. Kidneys are typically bean-shaped organs located near the back of the abdomen, just below the ribcage. They have a central cavity filled with urine, surrounded by renal tissue.

**3. Comparison with Known Anatomy:** In this image, there is a central dark area that resembles the renal pelvis, which is the main collecting duct system of the kidney. Surrounding this area, there appears to be a layer of tissue that could represent the renal cortex and medulla, which are the outer and inner layers of the kidney, respectively.

**4. Contrast and Tissue Differentiation:** The image shows a clear differentiation between the central dark area and the surrounding tissue, which is characteristic of the renal pelvis and the surrounding renal parenchyma. The presence of these distinct layers further supports the identification of the organ as a kidney.

**5. Conclusion Based on Evidence:** Given the anatomical features observed in the image—such as the central dark area resembling the renal pelvis and the surrounding tissue resembling the renal cortex and medulla—it is reasonable to conclude that the image indeed contains a kidney.

**Summary:** The image does contain a kidney based on the observed anatomical features and the differentiation between the renal pelvis and the surrounding renal parenchyma.

**Final Answer:** Yes.

Figure 7: Sample from our medical reasoning dataset: A **closed-ended** analysis of a **radiology** case, illustrating how our model breaks down diagnostic imaging through step-by-step clinical reasoning.

## C.4 CONFIDENCE DISTRIBUTION SHIFT

Figure 8 visualizes the impact of our training framework on the CARE model's confidence distribution across the VQA-RAD, SLAKE, and PathVQA datasets. The charts compare the distribution of confidence scores before training (red bars) and after training (green bars).

A clear and consistent trend is observable across all three datasets. Before training, the model's confidence is widely dispersed, with a significant number of predictions having low to moderate confidence scores. After training with our confidence-aware reward mechanism, there is a distinct shift in the distribution toward higher confidence levels. The green bars are predominantly concentrated on the right side of the charts, particularly in the 0.75 to 1.00 range, indicating that the model has

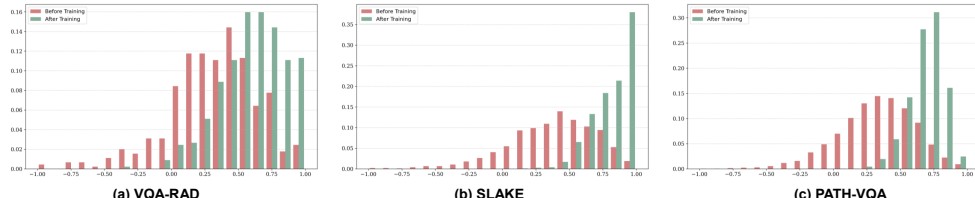

Figure 8: The change in **confidence distribution** of **CARE** before and after training on three datasets (after normalization), where red represents before-training and green represents after-training. It can be observed that the confidence-aware training framework not only improves answer accuracy but also significantly enhances confidence levels.

become significantly more confident in its predictions. This result demonstrates that our framework is effective not only in improving answer accuracy but also in enhancing the model's calibration, making it more certain of its correct responses.

