# OpenReview forum: "CARE: Confidence-Aware REasoning for Reliable Medical-VQA"
_ICLR.cc/2026/Conference — ICLR 2026 Conference Withdrawn Submission_

### Official Review · Reviewer_aZVc · 2025-10-21

**Soundness:** 2
**Presentation:** 2
**Contribution:** 2
**Rating:** 2
**Confidence:** 3

**Summary:**

This paper proposes CARE, an MLLM that incorporates Chain-Of-Thought (CoT) reasoning and confidence awareness into its training for medical VQA. CARE first generates medical CoT data using the base model and leverages GPT-4o as a verifier to ensure the conclusion aligns with the ground-truth answer. Then, it performs SFT on the CoT data, followed by an RL phase, which introduces its core design - confidence-aware reward (CAR). CAR takes the averaged probability of a response as the model's confidence and tries to align it with the response's accuracy.

**Strengths:**

1. Introducing confidence-aware reward (CAR) is beneficial to improving the calibration and reliability of medical MLLMs.
2. Experimental results show CARE effectively enhances accuracy and model calibration on the medical VQA task.

**Weaknesses:**

1. Limited contribution. The idea of SFT on CoT data and GRPO-based RL training has been widely adopted in medical MLLMs, and the paper’s core novelty lies solely in its confidence-aware reward design, which by itself is incremental.
2. Unclear generalization. It is unclear whether CARE's effectiveness generalizes to other medical tasks, e.g., report generation. Evaluating only on the medical VQA task is insufficient.
3. Marginal performance gain. It can be observed that CARE only achieves marginal performance gain on VQA-RAD, SLAKE, and PathVQA benchmarks compared to Llava-Med-7B, which is rather an outdated and weak baseline.

**Questions:**

Please see Weaknesses.

---

### Official Review · Reviewer_Kz3u · 2025-10-27

**Soundness:** 2
**Presentation:** 2
**Contribution:** 2
**Rating:** 2
**Confidence:** 5

**Summary:**

This paper introduces CARE, a Multimodal Large Language Model (MLLM) designed to improve the reliability of medical Visual Question Answering (VQA). The authors identify that existing models, even those trained with reinforcement learning, suffer from insufficient reliability and overconfidence, which is a major barrier to clinical use.

The CARE framework consists of two main stages:
1.  **Supervised Fine-Tuning (SFT):** The model is first trained on a newly constructed high-quality "Medical-CoT" (Chain-of-Thought) dataset. This dataset is generated by using a base model to produce reasoning steps that lead to a known ground-truth answer, which are then verified by an auxiliary model (GPT-4O).
2.  **Reinforcement Fine-Tuning (RFT):** The model is then fine-tuned using the Group Relative Policy Optimization (GRPO) algorithm. The key innovation is a novel "Confidence-Aware Reward (CAR)" mechanism. This reward function combines standard format and accuracy rewards with a calibration reward, which is designed to penalize the model for being highly confident in wrong answers and reward it for being confident in correct answers.

The authors evaluate CARE (using 3B and 7B backbones) on four medical VQA benchmarks (VQA-RAD, SLAKE, PathVQA, and OmniMedVQA) and show that it outperforms existing non-reasoning and reasoning-based medical MLLMs in both accuracy and reliability, as measured by Expected Calibration Error (ECE).

**Strengths:**

The paper rightly emphasizes that accuracy alone is insufficient for clinical adoption. By focusing on reliability and using Expected Calibration Error (ECE) as a primary evaluation metric, the paper addresses a crucial gap in current medical MLLM research.

**Weaknesses:**

1.  **Questionable RFT Training Configuration:** The most significant weakness is the RFT setup. As stated in Section 4.2, the use of a batch size of 2 and only 4 rollouts per sample is highly unconventional for a GRPO algorithm. This small group size is likely insufficient for computing a stable advantage function, which is critical for policy optimization. This setup would be prone to high variance and sparse rewards, which casts doubt on the stability and reproducibility of the training process and the robustness of the final results.
2.  **Outdated Baselines:** The paper compares CARE against baselines like Med-R1 and MedVLM-R1. However, it fails to include more recent and powerful state-of-the-art open MLLMs, such as newer versions of the Qwen-VL or InternVL series, as direct baselines. While a Qwen2.5 model is used as the backbone, its performance in a standard SFT setting is not compared, making the SOTA claims less convincing.
3.  **Limited Evaluation Benchmarks:** The evaluation is performed on VQA-RAD, SLAKE, PathVQA, and OmniMedVQA. While these are established datasets, the paper omits other, more recent and challenging medical reasoning benchmarks (e.g., MedXpertQA). Evaluating on these "simple VQA" datasets may not fully demonstrate the advanced reasoning and reliability capabilities claimed by the paper.

**Questions:**

1.  Could you please provide a detailed justification for the RFT training configuration? Specifically, how did you achieve stable training and convergence with a batch size of 2 and a group size of 4 for the GRPO algorithm? This setup seems insufficient for a reliable advantage calculation and would likely suffer from high gradient variance.
2.  Why were more recent state-of-the-art MLLMs, such as the InternVL 2.5/3 series, not included as baselines in Table 1?
3.  Why was a more challenging reasoning benchmark, such as MedXpertQA, not included in the evaluation?

---

### Official Review · Reviewer_okHL · 2025-11-01

**Soundness:** 3
**Presentation:** 3
**Contribution:** 3
**Rating:** 4
**Confidence:** 4

**Summary:**

The paper presents CARE, a confidence-aware reasoning method for Medical-VQA. The method pairs Chain-of-Thought training with a two-stage pipeline. The first stage uses supervised fine-tuning to teach step-by-step reasoning. The second stage uses reinforcement fine-tuning with a confidence-aware reward that joins answer accuracy and a calibration signal. The design includes custom data construction and a reward calibration scheme. The study treats both accuracy and reliability as first-class goals and reports expected calibration error as a key measure. Experiments on four Medical-VQA benchmarks show higher answer accuracy and lower calibration error than strong baselines.

**Strengths:**

1. CARE combines structured Chain-of-Thought reasoning with confidence calibration inside an RL setup. The explicit reward that includes confidence, described in Section 3.4 and shown in Figure 2, is clearly defined and easy to implement. It pushes the model toward reliable outputs rather than only higher accuracy.

2. The analysis of reasoning transfer is thoughtful. The Bayesian view in Section 4.2 and the study of interference to transfer in Section 4.3 help explain why general reasoning training can help in medical tasks. The paper connects theory and measurements in a clear way.

3. The evaluation covers both correctness and reliability. Reporting accuracy together with expected calibration error on multiple Medical-VQA datasets gives a more complete view of model behavior in a high-stakes setting. The breakdowns across datasets and metrics help readers see where the gains come from.

**Weaknesses:**

1. It is hard to credit the gains of CARE-7B over LLaVA-Med-7B to the proposed method because the setup is not controlled. CARE-7B uses a different and larger training set and a different vision-language backbone. Any gap could come from data scale or base model capacity rather than the confidence-aware reward or the RL stage. A fairness study would help. That study should retrain both methods on the same dataset and the same base model, and include ablations that isolate the effects of data, backbone, and training algorithm.


2. The paper does not position its use of confidence signals in RL rewards clearly against earlier work. Prior studies[1, 2, 3] already add confidence to reward design in reasoning systems. The paper cites and discusses these only briefly and does not run direct comparisons. As a result, the contribution reads as an incremental mix of CoT supervised fine-tuning plus GRPO-style training with a simple confidence term, rather than a clear step forward with demonstrated advantages.

3. The way confidence is defined may be off. The method treats confidence as the average token probability of the generated text and optimizes this value inside the reward as written in Equations 7 and 8. This proxy depends on length and phrasing and tracks how predictable the string is, not how uncertain the model is about the answer. It can be gamed by shorter or stock text and may reward smooth but shallow reasoning.



[1] Prabhudesai, Mihir, et al. "Maximizing Confidence Alone Improves Reasoning." *arXiv preprint arXiv:2505.22660* (2025).

[2] Zhao, Xuandong, et al. "Learning to reason without external rewards." *arXiv preprint arXiv:2505.19590* (2025).

[3] Fu, Yichao, et al. "Deep think with confidence." *arXiv preprint arXiv:2508.15260* (2025).

**Questions:**

Please see the weakness above.

---

### Official Review · Reviewer_5d11 · 2025-11-10

**Soundness:** 3
**Presentation:** 4
**Contribution:** 2
**Rating:** 4
**Confidence:** 4

**Summary:**

The paper proposes a two-stage training framework, CARE, for medical VQA in multimodal LLMs. Stage-1 uses structured CoT-style SFT to elicit step-wise diagnostic reasoning; Stage-2 applies GRPO-based reinforcement fine-tuning with a Confidence-Aware Reward that combines format compliance, an output score, and a calibration term computed as the average token probability across the generated sequence (reasoning + answer). The authors claim CARE improves both accuracy and Expected Calibration Error (ECE) across VQA-RAD, SLAKE, PathVQA, and OmniMedVQA.

**Strengths:**

1. The motivation is clear and reasonable.
2. The experiments on four datasets are extensive.
3. The paper presents SOTA performances.

**Weaknesses:**

1. The ideas of confidence-aware medical LLMs and confidence with RL have been explored in previous works [Refs. 1–5].

2. Several important core contributions and claims are not supported.
- The paper claims that the model can generate reliable and trustworthy reasoning.
- The claimed robustness and generalization ability of the proposed method are also not validated. For example, the generalization ability on external datasets should be verified.
- All validation focuses on answers rather than reasoning validity. All results are obtained on existing VQA benchmarks; there is no reader study with clinicians, no prospective test, and no robust OOD stress testing.

3. GPT-4o accepts a CoT if the final conclusion aligns perfectly with the ground truth; there is no explicit check for hallucinated or clinically invalid intermediate steps. This weakens the claim of trustworthy reasoning.

4. Baselines and fairness of comparisons are unclear. It is not clear whether non-reasoning baselines were retrained under identical data and prompts, or whether all reasoning baselines also received comparable process supervision and GRPO budgets/settings. This matters for the closed-set gains on OmniMedVQA and PathVQA.

5. The confidence proxy is fragile and length-biased. Confidence is computed as the average token probability over the entire output sequence (reasoning + answer). Longer or stylistically verbose traces can lower mean probability irrespective of correctness; per-token probability is also temperature-dependent. No normalization for length or separation between reasoning and answer tokens is described.

6. Open-ended scoring (“recall”) is under-specified. Eq. (6) defines recall(o_i, GT) without operational details (n-gram level? synonyms? stemming? clinical terminology variants?). Longer outputs can artificially inflate recall, conflating verbosity with correctness; no human evaluation of free-text factuality is provided.

7. Data transparency for the generated Medical-CoT set is insufficient. The paper does not report the size, modality distribution, or sampling strategy of the synthesized CoTs, nor the verifier acceptance rate.

8. The checklist marks “Yes” for formal theorems/proofs, which do not appear in the paper.

Refs.

[1] Probabilistic medical predictions of large language models. In npj Digital Medicine, 2024.

[2] Benchmarking the Confidence of Large Language Models in Answering Clinical Questions: Cross-Sectional Evaluation Study. In JMIR Medical Informatics, 2025.

[3] Confidence Calibration for Multimodal LLMs: An Empirical Study through Medical VQA. In MICCAI, 2025.

[4] Prompt4Trust: A Reinforcement Learning Prompt Augmentation Framework for Clinically-Aligned Confidence Calibration in Multimodal Large Language Models. In ICCV 2025

[5] Confidence Is All You Need: Few-Shot RL Fine-Tuning of Language Models. In ArXiv Preprint, 2025.

**Questions:**

1. The ideas of confidence-aware medical LLMs and confidence with RL have been explored in previous works [Refs. 1–5].

2. Several important core contributions and claims are not supported.
- The paper claims that the model can generate reliable and trustworthy reasoning.
- The claimed robustness and generalization ability of the proposed method are also not validated. For example, the generalization ability on external datasets should be verified.
- All validation focuses on answers rather than reasoning validity. All results are obtained on existing VQA benchmarks; there is no reader study with clinicians, no prospective test, and no robust OOD stress testing.

3. GPT-4o accepts a CoT if the final conclusion aligns perfectly with the ground truth; there is no explicit check for hallucinated or clinically invalid intermediate steps. This weakens the claim of trustworthy reasoning.

4. Baselines and fairness of comparisons are unclear. It is not clear whether non-reasoning baselines were retrained under identical data and prompts, or whether all reasoning baselines also received comparable process supervision and GRPO budgets/settings. This matters for the closed-set gains on OmniMedVQA and PathVQA.

5. The confidence proxy is fragile and length-biased. Confidence is computed as the average token probability over the entire output sequence (reasoning + answer). Longer or stylistically verbose traces can lower mean probability irrespective of correctness; per-token probability is also temperature-dependent. No normalization for length or separation between reasoning and answer tokens is described.

6. Open-ended scoring (“recall”) is under-specified. Eq. (6) defines recall(o_i, GT) without operational details (n-gram level? synonyms? stemming? clinical terminology variants?). Longer outputs can artificially inflate recall, conflating verbosity with correctness; no human evaluation of free-text factuality is provided.

7. Data transparency for the generated Medical-CoT set is insufficient. The paper does not report the size, modality distribution, or sampling strategy of the synthesized CoTs, nor the verifier acceptance rate.

8. The checklist marks “Yes” for formal theorems/proofs, which do not appear in the paper.

---

### Note · Authors · 2025-11-14

I have read and agree with the venue's withdrawal policy on behalf of myself and my co-authors.